# Use of a Practitioner-Friendly Behavior Model to Identify Factors Associated with COVID-19 Vaccination and Other Behaviors

**DOI:** 10.3390/vaccines10081261

**Published:** 2022-08-05

**Authors:** Sohail Agha

**Affiliations:** Behavior Design Lab, Stanford University, Stanford, CA 94305, USA; sohailagha@gmail.com

**Keywords:** COVID-19, vaccination, vaccine uptake, condom use, iron folate use, contraceptive use, behavior change, behavior model, behavior design, Fogg Behavior Model

## Abstract

The lack of capacity for the design and implementation of behavioral interventions in low-and-middle income countries (LMICs) has been recognized by the World Health Organization (WHO) and other global health institutions. There is a need to task-shift, to translate social and behavioral science concepts into “practitioner-friendly” models—models which can be used by intervention designers, implementers, and evaluators with limited technical and financial resources. We illustrate the use of the Fogg Behavior Model (FBM), a model identified as being easy for practitioners to adopt in low-resource settings. The study uses data across four different behaviors in Nigeria, Pakistan, and India. The behaviors examined are COVID-19 vaccine uptake, condom use, iron folate use, and modern contraceptive use. The data are from surveys of healthcare workers (HCWs), married men, women of reproductive age, and adolescents, respectively. The FBM states that behavior happens when both motivation and ability are present, and a prompt occurs. In other words, persons with high motivation and high ability are the first to adopt a behavior. We created a categorical variable for motivation and ability and tested whether high motivation and high ability are associated with a greater likelihood of adopting a behavior. In Nigeria, HCWs with high motivation and high ability had 27 times higher odds of being vaccinated. In Pakistan, married men with high motivation and high ability had 35 times higher odds of condom use with their wives. In India, women with high motivation and high ability had 9 times higher odds of iron folate use. In Nigeria, adolescents and young women with high motivation and high ability had 8 times higher odds of contraceptive use. The study findings suggest that the FBM has the potential to be applied in low resource settings for the design, implementation, and evaluation of behavioral interventions. Rigorous testing of the FBM using data from experimental or quasi-experimental studies is recommended.

## 1. Introduction

Evidence suggests that theory-based behavioral interventions are more likely to have an impact [1], yet few behavioral interventions in low-and-middle income countries (LMICs) are explicitly designed based on a theory of behavior. The increase in the number of theories of behavior—estimated to be more than 80 [2]—has not led to an accompanying increase in their adoption by practitioners. The complexity of these theories and the challenges faced by practitioners in understanding them have been discussed elsewhere [3,4]. Recognizing that the application of behavior change theories by behavior change practitioners working in LMICs remains very limited, UNICEF has called for making behavior models more practical and easy to use [5]. Without a good understanding of the behavioral drivers involved, practitioners risk resorting to default interventions, such as those trying to raise awareness, which may not address the problem at hand [5]. A recent systematic review of behavior change theories implemented in LMICs has also highlighted the fact that, in the context of implementation of development programs, theoretical complexity becomes a barrier to successful practice [6].

The lack of formal training opportunities for practitioners in social and behavior change (SBC) in LMICs compounds the problem. Funding constraints for behavioral interventions are an additional barrier. There is growing recognition among donors and multilaterals of the need to increase the capacity for behavior change programming among practitioners in LMICs. Donors such as the Bill & Melinda Gates Foundation have initiated investments to increase the capacity of a broad spectrum of practitioners and researchers based in Ghana and Nepal to understand and apply behavior change theories [7]. A recent survey performed among immunization professionals in LMICs showed that only 2 out of 48 respondents from ministries of health, international NGOs, and private sector organizations felt that they understood behavioral science models [8]. In 2020, the WHO initiated efforts to increase the use of behavioral sciences across LMICs by establishing a Behavioral Insights unit in its head office. The WHO is now building capacity for the behavioral sciences by providing technical assistance to countries through its regional offices and by hiring behavior change experts. One of the key areas of focus of this capacity building is “building skills in behavioral theory and frameworks” [9]. UNICEF recognizes that capacity for behavior change programming in many countries remains below par even though some efforts have been made to increase capacity in this area [10]. A recent evaluation has emphasized the need for increasing the quality of behavior change capacity building efforts in LMICs [11]. As donors recognize the need to build an eco-system for behavioral science in LMICs, experts have emphasized the need to generate evidence for behavior change theories that are targeted to practitioners [6].

Given the lack of an eco-system that supports the design and implementation of behavioral interventions in LMICs, there is a need for “practitioner-friendly” models of behavior. These models simplify the more complex understanding of behavior that exists in the behavioral sciences into constructs that are intuitive to practitioners and are, ideally, jargon-free. The models should be parsimonious in terms of the number of constructs employed. Their application, as well as the research required to generate context-specific understandings of behavioral drivers, should be possible with limited resources. It should be possible to provide data for these models without an unduly high burden of data collection, which may result from surveys with long questionnaires. Ultimately, practitioner-friendly models should enable practitioners with limited training in the social and behavioral sciences to apply a behavioral lens to intervention design and implementation [5].

Enabling practitioners based in LMICs to use behavior models will have an additional benefit—it will enrich theories of behavior by capturing drivers of and barriers to behavior change that are more dominant in low resource settings. For instance, it took until 2017 for a major review of the immunization literature to identify “practical issues” such as knowing where to receive vaccination, ease of access, and affordability as barriers to the adoption of childhood immunization in LMICs [12]. Moreover, the lead researcher of this review noted that the participation of researchers who had themselves worked in LMICs led to “practical issues” being highlighted [13]. A review of behavioral interventions in immunization completed in 2022 confirms that immunization interventions have indeed not addressed ability-related barriers to vaccine uptake [14].

A systematic review of the use of behavior change theories in LMICs shows a gradual increase in their use since 2015 [6]. The review called for more evidence to be generated in support of behavior models that are easier for practitioners to use. In this study, we illustrate the use of a model that has begun to be used in LMICs, the Fogg Behavior Model (FBM) [15]. This study generates evidence for the FBM in multiple social and behavioral contexts.

The FBM can be visualized in two dimensions, with motivation on the y-axis and ability on the x-axis—as shown in Figure 1. For a specific behavior, motivation can range from high to low. Ability can also range from high to low for a particular behavior. For simplicity, we say a behavior is easy to do or hard to do. The FBM proposes that a behavior happens when a person with high motivation and high ability is prompted. By contrast, a person with low motivation and low ability is not likely to adopt a behavior when prompted. Fogg considers a threshold (the “action line”) above which a person with sufficient motivation and ability will adopt a behavior when prompted [15,16]. We investigate whether, independent of other factors, an individual with high motivation and high ability is significantly more likely to adopt a behavior than one with lower levels of motivation and ability.

## 2. Materials and Methods

We tested the utility of the FBM for both formative research studies (in the case of HCWs and adolescents and young women) as well as evaluations (in the case of married men and reproductive age women). It is important to use behavior change theory in the design of formative research conducted to develop interventions [17]. This prevents key drivers of behavior from being left out inadvertently and ensures that interventions are developed in line with behavioral theory [18]. Similarly, the importance of evaluations being based on theory has been highlighted. Conducting theory-based evaluations of behavioral interventions is important for understanding the mechanisms through which effects operate and ensures that the interpretation of findings is consistent with theory.

### 2.1. Survey Data

#### 2.1.1. Nigerian Healthcare Workers (HCWs)

In July 2021, an online survey of Nigerian healthcare workers (HCWs) was conducted to identify factors associated with the uptake of a COVID-19 vaccine. The survey was conducted with HCWs in all six regions of Nigeria. Respondents were recruited via advertisements on Facebook using the Virtual Lab open-source tool [19]. Recruitment was stratified by age, region, education, and occupation. Respondents were asked about their age, gender, level of education, what type of provider they were, how well they thought the government was managing the COVID-19 pandemic, and their vaccination status. Respondents were asked one question about their motivation and another about their ability to get vaccinated.

At the time the survey was conducted, HCWs had been the focus of the National Primary Healthcare Development Agency’s (NPHCDA) vaccination efforts. Of the 2364 people who clicked on the ad, 697 gave consent and 496 completed the survey. More details of the survey methodology are provided elsewhere [20]. The behavior of interest was an HCW getting at least one dose of a COVID-19 vaccine.

#### 2.1.2. Pakistani Men

In 2009, a household panel survey of men married to women aged 15–49 was conducted in urban Pakistan to assess the effects of a social marketing condom promotion campaign. The survey was conducted in four provinces of Pakistan: Sindh, Punjab, Khyber-Pakhtunkhwa, and Baluchistan. The instrument used for the survey included questions on the socio-economic and demographic characteristics of respondents, their attitudes towards condoms and family planning, and their exposure to a condom advertising campaign.

The campaign was implemented in two phases. Phase 1 of the campaign was implemented in February–March 2009. In total, 2156 advertisements were aired on television during Phase 1. The first wave of data was collected in March–April 2009 to monitor the campaign’s effects. Phase 2 of the campaign was implemented in April–May 2009, and 2311 advertisements were aired. The second wave of data was collected in August 2009. A multistage, cluster, random sampling strategy was used to collect the data. Details of the sample design are provided elsewhere [21]. A total of 806 men were interviewed in the first survey wave, and 617 were re-interviewed in the second wave. We used the panel of 617 married men to test the applicability of the FBM to the behavior of interest—a man’s use of a condom during last sex with his wife.

#### 2.1.3. Indian Women

In 2020, an online survey was conducted among women aged 18–49 in the states of Uttar Pradesh and Madhya Pradesh in India. The survey was conducted using a combination of the Typeform survey tool and Facebook advertisements. It asked women about their age, education, iron folate use and exposure to Facebook posts on anemia and iron folate. Women were also asked about their motivation and ability to use iron folate. Details of the survey methodology are provided elsewhere [22,23]. In 2019, the year before the survey, two online advertising campaigns were conducted on Facebook to raise awareness of anemia and to increase the use of iron folate among women of reproductive age in these two states [22]. Of the 1365 women interviewed, 1136 provided responses to all seven questions asked in the survey and were included in the analysis. The behavior of interest was a woman’s recent use of iron folate.

#### 2.1.4. Nigerian Adolescents and Young Women

In 2018, a household survey of women of ages 14–24 was conducted in Lagos, Kaduna and Kano states in Nigeria. The survey provided baseline data on attitudes, beliefs, practices, and socio-economic and demographic characteristics of women. Data were collected in February–March, using a multistage, cluster, random sample. Details of the sample design are provided elsewhere [24]. In total, 2890 women were interviewed in the three states. We tested the FBM using data about 618 Nigerian women who reported ever having had sex. The behavior of interest was a woman’s current use of a modern contraceptive method.

### 2.2. Operationalization of the Fogg Behavior Model

In Nigeria, HCW motivation and ability were measured by asking a global question to measure each construct. To measure motivation, Nigerian HCWs were asked “How important do you think getting the COVID-19 vaccine is for your health?” Responses were recorded on a Likert scale, which ranged from “not at all important” to “very important”. A binary variable was created for motivation, with HCWs who reported that it was very important to get vaccinated coded as 1 and other HCWs coded as 0. Responses to the ability question “How easy or difficult is it for you to get a COVID-19 vaccination for yourself” were also recorded on a Likert scale. A binary variable was created for ability, with respondents who found it very easy to get vaccinated coded as 1 and the remaining respondents coded as 0.

A concern with converting a variable on a Likert scale to a binary variable is that the relationship between the converted variable and other variables will change [25]. We tested whether this had occurred. First, we looked at the relationship between the two independent variables, motivation and ability. The correlation between these two variables before dichotomization was r = 0.221 (r^2^ = 0.049). The correlation between these two variables after dichotomization was r = 0.218 (r^2^ = 0.048). Second, we looked at the relationship between each of the independent variables and the behavioral outcome, receiving at least one COVID-19 vaccination. The correlation between the outcome and motivation was r = 0.334 (r^2^ = 0.112) before the dichotomization. The correlation between motivation and the outcome was r=0.343 (r^2^ = 0.118) after the dichotomization. Before the dichotomization, the correlation between the outcome and ability was r = 0.506 (r^2^ = 0.256). After dichotomization, the correlation between the outcome and ability was r=0.429 (r^2^ = 0.184). As shown, above, there was no change in our substantive findings and interpretation after the conversion of the Likert scale variables to dichotomous variables. Thus, converting the motivation and ability variables from the Likert scale to dichotomous variables did not create a bias.

A cross tabulation between the motivation and ability binary variables provided four possible combinations of motivation and ability: high motivation and high ability, high motivation and low ability, low motivation and high ability, and low motivation and low ability. Bivariate analysis showed that HCWs with high motivation and low ability had similar levels of vaccination as HCWs with low motivation and high ability. These two categories were combined to obtain a three-category motivation-ability variable: high motivation and high ability, high motivation or high ability, and low motivation and low ability.

In Pakistan, a series of questions were asked to measure motivation and ability amongst married men. The first half of Table 1 shows that 14 questions were asked to measure motivation. The FBM defines motivation as comprising anticipation (i.e., hope or fear), sensation (i.e., pleasure or pain) and belonging (i.e., social acceptance or rejection) [15]. The second half of Table 1 shows that 15 questions were asked to measure ability. The FBM defines ability as comprising time, money, physical effort, mental effort, and routine [15].

Variables were coded in a positive direction in relation to the behavioral outcome of interest. Each survey question with a positive response that reflected motivation or ability was given a score of 1. This meant that motivation had a maximum score of 14, while ability had a maximum score of 15. The upper quartiles of each variable were used to define “high motivation” and “high ability”. A cross tabulation was performed between high motivation and high ability, with four categories emerging: high motivation and high ability, high motivation and low ability, low motivation and high ability, and low motivation and low ability. Bivariate analysis showed that men with high motivation and low ability had a similar level of condom use as men with low motivation and high ability. These two categories were collapsed into one, resulting in a three-category motivation-ability variable: high motivation and high ability, high motivation or high ability, and low motivation and low ability.

In India, motivation and ability were measured by asking a global question to measure each construct. To measure motivation, women were asked “How important is it for you, personally, to take iron supplements?” Their responses were recorded on a Likert scale that ranged from very important to not at all important. A binary variable was created for motivation, with respondents who reported that it was very important for them to take an iron supplement coded as 1 and other respondents coded as 0. To measure ability, women were asked “If you decided to take an iron supplement, how difficult would it be for you to obtain iron tablets in the next week?” Responses were recorded on a Likert scale from very easy to very difficult. A binary variable was created for ability, with respondents who felt that it would be very easy for them to obtain an iron supplement coded as 1 and other respondents coded as 0.

Cross tabulations between the two binary variables provided four possible combinations of motivation and ability. Following the bivariate analysis, which showed that individuals with high motivation and low ability and those with low motivation and high ability had a similar level of iron folate use, we combined these two categories. This resulted in a three-category motivation-ability variable: high motivation and high ability, high motivation or high ability, and low motivation and low ability.

For adolescents and young women in Nigeria, Table 2 shows the survey questions used to operationalize motivation and ability. The survey asked 23 questions to operationalize motivation and 18 to operationalize ability. Variables were coded in a positive direction in relation to the behavioral outcome of interest. Each survey question that reflected motivation or ability was given a score of 1. As a result, motivation had a maximum score of 23, while ability had a maximum score of 18. The upper quartiles of each variable were used to define the binary variables of “high motivation” and “high ability”.

Cross tabulation between these two binary variables provided four possible combinations of motivation and ability. As with the data for HCWs in Nigeria, married men in Pakistan and reproductive-age women in India, we conducted bivariate analysis to examine the relationship between the four combinations of motivation and ability and modern contraceptive use. This showed that the categories of high motivation and low ability and low motivation and high ability had similar levels of modern contraceptive use. Following this analysis, we combined the category of high motivation and low ability with the category of low motivation and high ability. This resulted in a three-category motivation-ability variable: high motivation and high ability, high motivation or high ability, and low motivation and low ability.

### 2.3. Statistical Analysis

Frequency distributions of the socio-economic and demographic characteristics of respondents and their motivation and ability are shown in Table 3, Table 4, Table 5 and Table 6. Cross tabulations between variables and the behavior of interest are also shown in these tables. We tested the following hypotheses:(1)Respondents with high motivation and high ability are more likely to adopt a behavior than respondents with low motivation and low ability.(2)Respondents with high motivation or high ability are more likely to adopt a behavior than respondents with low motivation and low ability.(3)Respondents with high motivation and high ability are more likely to adopt a behavior than respondents with high motivation or high ability.

Multivariate analysis was used to test the hypotheses. The clustering of observations (i.e., the clustering of respondents in sampling units) was taken into account by using the STATA cluster command [26]. For the Pakistan panel, we used a multi-level mixed effects logistic regression [26] to take the clustering of observations within cities and individuals into account.

### 2.4. Study Limitations

A limitation of this study is the use of existing data from two surveys (i.e., the survey of men in Pakistan and the survey of adolescents and young women in Nigeria) that were not specifically designed to collect data on constructs articulated by the FBM. As a result, several important elements that comprise ability in the FBM were not measured in the Pakistan (i.e., time, routine) and Nigeria (i.e., time, money, physical effort) surveys. Another limitation of this study is the reliance on self-reported behavior, which may be influenced by social desirability or recall bias [27]. Finally, because we used cross-sectional survey data in three out of four cases, no causal inferences can be drawn from the analysis.

We used global, single-item scales of motivation and ability in two of the studies: the study with HCWs in Nigeria and the study with women of reproductive age in India. The use of multi-item scales is recommended in the literature and widely practiced by researchers. While there are concerns regarding measurement error resulting from single-item scales, several studies have shown that single-item scales perform as well or better for topics such as job satisfaction [28] or perform reasonably well for topics such as quality of life [29] or depression [30]. More research is needed to determine how well single-item measures of motivation and ability perform against multi-item scales.

### 2.5. Ethics Review

All protocols were carried out in accordance with relevant guidelines and regulations. Informed consent was obtained from study participants. For married men in Pakistan and adolescents and young women in Nigeria, Tulane University Biomedical IRB approval was received. The IRB reference number for the Pakistan study is 09-141661. The IRB reference number for the Nigeria study is 2017-6388. Local IRB approvals were obtained from the Pakistan Medical Association and the Nigerian National Health Research Committee. The IRB approval for India was obtained from the University of California San Francisco IRB—IRB reference number 18-27024. For the study on HCWs in Nigeria, the University of Washington Human Subjects Division reviewed the protocol and made a “Determination of Exempt Status”.

## 3. Results

Column 1 of Table 3 shows frequency distributions of the three-category motivation-ability variable as well as characteristics of Nigerian HCWs. Only 27% of HCWs had high motivation and high ability to obtain a COVID-19 vaccination. Another 47% of HCWs had either high motivation or high ability to get vaccinated, while the remaining 26% had low motivation and low ability.

The survey sample comprised HCWs from all six regions of Nigeria, with the largest proportion being from the South West. Slightly more than half of HCWs were between the ages of 30 and 39. Nearly half of HCWs were women. Most HCWs had bachelor’s or higher education. Physicians comprised 16% of the sample. The majority of HCWs felt that the National Primary Health Care Development Agency (NPHCDA) was managing the COVID-19 pandemic well.

Column 2 of Table 3 shows cross tabulations between independent variables and an HCW’s getting at least one dose of a COVID-19 vaccine. There was a powerful bivariate association between motivation, ability, and getting vaccinated: 89% of HCWs with high motivation and high ability had gotten at least one dose of a COVID-19 vaccine, compared to 56% with high motivation or high ability, and 25% with low motivation and low ability.

Vaccine uptake varied by region, with vaccination levels being highest in the North West (75%) and lowest in the South South (39%). Physicians reported a higher level of vaccine uptake (70%) than other HCWs (54%). Vaccine uptake was higher among those who believed that the NPHCDA had managed COVID-19 somewhat (56%) or definitely well (65%), compared to those who did not believe this or did not know (42%).

Column 3 of Table 3 shows the adjusted odds of a Nigerian HCW getting at least one dose of a COVID-19 vaccine. An HCW with high motivation and high ability had 27 times higher odds of being vaccinated compared with an HCW with low motivation and low ability (aOR = 26.6, *p* < 0.001). This finding provides support for our first hypothesis. An HCW with high motivation or high ability had four times higher odds of being vaccinated compared with an HCW with low motivation and low ability (aOR = 3.94, *p* < 0.001). This finding provides support for our second hypothesis.

In a separate logistic regression analysis (not shown), we used high motivation or high ability as the reference category. We found that an HCW with high motivation and high ability had six times higher odds of being vaccinated compared with an HCW with high motivation or high ability (aOR = 5.64, *p* < 0.001). This finding provides support for our third hypothesis. Overall, these findings are consistent with three levels of effects of the motivation and ability variable on vaccine uptake by HCWs.

In terms of the relationships between other variables and getting vaccinated, HCWs in the North West (aOR = 3.97, *p* < 0.01) and HCWs in North Central (aOR = 2.23, *p* < 0.05) were more likely to get vaccinated than HCWs in the South South. Physicians were more likely to get vaccinated than other HCWs (aOR = 2.68, *p* < 0.01). HCWs who believed that the National Primary Health Care Development Agency was managing COVID-19 well had about twice the odds of obtaining at least one dose of a COVID-19 vaccine.

Column 1 of Table 4 shows frequency distributions of characteristics of married Pakistani men. Only 19% of men had high motivation and high ability to use condoms. About 25% of men had high motivation or high ability, while the majority (56%) had low motivation and low ability to use condoms.

Nearly half the men in the sample were ages 40 and above. About half had four or more children. A substantial proportion of married men (43%) did not want additional children. Slightly more than half the men had grade 10 or higher education. About 15% of men had been exposed to a condom social marketing campaign.

Column 2 of Table 4 shows cross tabulations between motivation and ability and a married man’s use of a condom during last sex with his wife, at baseline. At the bivariate level, there was a strong relationship between a man’s motivation and ability and condom use: 44% of men with high motivation and high ability used a condom during last sex, compared with 25% of men with high motivation or high ability, and 7% of men with low motivation and low ability.

Condom use was higher among men who wanted no additional children compared with men who did (26% versus 13%). Men in the wealthiest quintile were more likely to use condoms than men in the other 4 quintiles (25% versus 17%). Exposure to condom advertising was associated with higher condom use (28% versus 17%). Overall, condom use increased from 18.5% to 22% between baseline and follow-up (not shown). While condom use increased between baseline and follow-up, the relationships between independent variables and condom use remained similar at follow-up (not shown).

Column 3 of Table 4 shows the adjusted odds of a married Pakistani man using a condom during last sex with his wife. A man with high motivation and high ability had 35 times higher odds of condom use than a man with low motivation and low ability (aOR = 35.33, *p* < 0.001). This finding provides support for Hypothesis 1. A man with high motivation or high ability also had six times higher odds of condom use compared with a man with low motivation and low ability (aOR = 6.36, *p* < 0.001). This finding provides support for Hypothesis 2.

In a separate regression analysis (not shown), we used high motivation or high ability as the reference category. We found that a man with high motivation and high ability had six times higher odds of condom use compared with a man with high motivation or high ability (aOR = 5.56, *p* < 0.001). This finding provides support for Hypothesis 3. Overall, these findings are consistent with the three levels of effects of the motivation and ability variable on the vaccination uptake behavior observed earlier.

In terms of the relationship between other variables and condom use, the odds of condom use were higher among men with 2–3 children and among men who wanted no additional children. Men who were exposed to condom advertising had higher odds of condom use. The increase in condom use wave 1 and wave 2 surveys did not remain significant after adjusting for exposure to advertising.

Column 1 of Table 5 shows frequency distributions of characteristics of Indian women in the sample. Only 16% of women had high motivation and high ability to use iron folate. Another 31% of women had high motivation or high ability, while the majority had low motivation and low ability to use iron folate (53%).

Slightly more than half the women in the sample were less than 26 years of age. One-fifth of women had less than 5 years of schooling. About 16% of women had been exposed to a social media campaign on anemia prevention.

Column 2 of Table 5 shows cross tabulations between independent variables and iron folate use. A strong relationship was observed between motivation and ability and iron folate use: 64% of women with high motivation and high ability reported use of iron folate, compared with 28% of women with high motivation or high ability, and 13% of women with low motivation and low ability.

Women aged 26 and older were more likely to use iron folate. Women with fewer than 5 years of schooling were more likely to use iron folate. Exposure to a social media campaign on anemia prevention was associated with higher iron folate use.

Column 3 of Table 5 shows the adjusted odds of iron folate use among Indian women. A significant relationship was observed between motivation, ability and iron folate use at the multivariate level: women with high motivation and high ability had nine times higher odds of using iron folate compared with women with low motivation and low ability (aOR = 9.41, *p* < 0.001). This finding supports Hypothesis 1. Women with high motivation or high ability had twice the odds of using iron folate compared with women with low motivation and low ability (aOR = 2.23, *p* < 0.001). This finding supports Hypothesis 2.

In a separate logistic regression analysis (not shown), we used high motivation or high ability as the reference category. We found that a woman with high motivation and high ability had four times higher odds of iron folate use compared with a woman with high motivation or high ability (aOR = 4.22, *p* < 0.001). This finding supports Hypothesis 3. Overall, these findings are consistent with the three levels of effects of motivation and ability observed for vaccination among HCWs and for condom use among married men.

Column 3 of Table 5 also shows relationships between women’s characteristics and iron folate use. After adjusting for other variables, women aged 26 and older were more likely to use iron folate. Women with less than 5 years of schooling were more likely to use iron folate. Women exposed to digital advertising promoting anemia prevention were also more likely to use iron folate.

Column 1 of Table 6 shows frequency distributions of characteristics of Nigerian adolescents and young women. Only 13% of young Nigerian women had high motivation and high ability to use modern contraception. Another 23% had high motivation or high ability, while the majority of adolescents and young women (64%) had low motivation and low ability to use modern contraception.

Nearly two-thirds of the survey sample was from Northern Nigeria. About three-fourths of women were aged 20–24, and 64% were married. Slightly more than half the women in the sample had at least one child. Most women had secondary or higher education.

Column 2 of Table 6 shows cross tabulations between independent variables and the use of modern contraception. There was a strong relationship between motivation, ability and the use of modern contraception at the bivariate level: 61% of women with high motivation and high ability were using a modern contraceptive, compared with 44% of women with high motivation or high ability and 14% of women with low motivation and low ability.

Contraceptive use was higher among women in the South, women who had a boyfriend, and women who did not have a child. Women who had higher than secondary education were more likely to use contraception. Women in the first/poorest quintile were less likely to use contraception compared to women in the second to fifth quintiles.

Column 3 of Table 6 shows the adjusted odds of modern contraceptive use among Nigerian adolescents and young women. A strong relationship between motivation, ability and modern contraceptive use was observed: compared with women with low motivation and low ability, women with high motivation and high ability (aOR = 7.91, *p* < 0.001) and women with high motivation or high ability (aOR = 3.93, *p* < 0.001) were more likely to use modern contraception. These findings provide support for Hypotheses 1 and 2, respectively.

In a separate logistic regression analysis (not shown), we used high motivation or high ability as the reference category and found that women with high motivation and high ability were significantly more likely to use modern contraception than women with high motivation or high ability (aOR = 2.01, *p* < 0.05). This finding provides support for Hypothesis 3. This pattern of three levels of effects of motivation and ability on behavior was also observed for vaccine uptake in Nigeria, for condom use in Pakistan, and for iron folate use in India.

Column 3 of Table 6 also shows multivariate relationships between characteristics of young Nigerian women and modern contraceptive use. Contraceptive use was higher among women who had a boyfriend and lower among women in the poorest quintile.

## 4. Discussion

This study illustrates the use of a practitioner-friendly behavior change model that has recently been introduced in the public health literature and is being used by practitioners in LMICs [6]. The COVID-19 pandemic laid bare the urgency to design behavioral interventions to increase COVID-19 vaccine uptake. At the same time, it has highlighted the absence of behavior change models in designing immunization interventions. There are a number of reasons why behavioral interventions are not being used to increase vaccine uptake in LMICs. These include limited funding for the design and implementation of behavioral interventions, practitioners’ lack of familiarity with and understanding of behavior change models, and limited resources for training practitioners. Given these constraints, there is a need to facilitate use of behavior models that are easy for practitioners to understand and implement. We believe that a practitioner-friendly behavior model should have a limited number of clearly articulated, intuitive, constructs that are easy to operationalize and collect data on.

The Fogg Behavior Model (FBM) has three constructs: motivation, ability, and a prompt. In the case of two of the behaviors examined in this study (condom use by married men and contraceptive use by young women), we operationalized motivation and ability constructs using data from surveys with longer questionnaires. These instruments included questions on the elements that comprise motivation (anticipation, sensation, belonging) and ability (time, money, physical effort, mental effort, routine). For the other two behaviors examined (COVID-19 vaccination and iron folate use), we operationalized motivation and ability constructs using data from surveys with brief instruments that included one global question on motivation and one global question on ability.

In explaining the adoption of these four behaviors, we found that a motivation-ability variable with three categories showed a powerful correlation with behavior. We found that motivation and ability had three distinct levels of effects: high motivation and high ability had the strongest effect, followed by high motivation or high ability and low motivation and low ability.

That three empirically meaningful segments based on motivation and ability can be obtained easily by asking one question on motivation and one on ability suggests that use of the FBM will bear immediate rewards for practitioners. It will allow practitioners to measure the size of each of these segments and decide how to use project resources to influence behavior. Use of the model will inform a practitioner about whether a behavioral intervention should focus on motivation, or ability, or on both motivation and ability. Depending on the size of each segment and the level of resourcing available for an intervention, an implementer may decide to focus on one or more segments simultaneously.

In the case of contraceptive use among young Nigerian women and married Pakistani men, our analysis showed that the majority of respondents had low motivation and low ability: 64% of young women in Nigeria and 56% of married men in Pakistan had low motivation and low ability to adopt contraception. Segments with low motivation and low ability are likely to require more project resources and time for behavior change to occur. In such an instance, an early win for implementers might be to focus on prompts to action for the segment with high motivation and high ability—a segment that is on the verge of behavioral adoption. Following an early win, implementers could focus on increasing motivation (for those with low motivation and high ability) or increasing ability (for those with high motivation and low ability). A staged approach with wins at each stage would help build confidence among practitioners as they take on the task of increasing motivation or increasing ability or increasing both motivation and ability.

Several limitations of our study should be kept in mind. Since cross-sectional data were used in three out of the four analyses conducted for this study, causal inferences cannot be made. Experimental data, ideally using a panel design, are needed to provide more robust empirical evidence that supports the use of the FBM. It is worth noting that, since high motivation and low ability and low motivation and high ability have a similar level of effect on behavior, we combined these two segments to form the high motivation or high ability segment. While combining these two segments made sense for this empirical analysis, the distinction between these two segments is important to maintain when designing interventions. For individuals with high motivation and low ability, interventions should focus on increasing ability and, for individuals with low motivation and high ability, interventions should focus on increasing motivation.

It is worth noting that the FBM is not *per se* a model of health behavior. It is a model of human behavior that we have applied to health behavior. The FBM has been applied to promoting lifestyle changes such as getting exercise or preparing healthy meals. For example, as obesity has risen in LMICs, application of behavior models to improved lifestyle behaviors has become increasingly important. As interest grows in applying behavioral insights to policy and advocacy interventions, to the adoption of financial and agricultural services, to increasing women’s participation in decision-making, and to climate change, a practitioner-friendly model that is applicable to a broad range of behaviors and in multiple social contexts becomes increasingly relevant.

Perhaps it is most useful to think of behavior change theories along a continuum, ranging from those that are very precise in predicting behaviors to those that can be applied by practitioners with limited resources and training in the social and behavioral sciences. There is a need to understand how well practitioner-friendly models of behavior perform against more traditional models that have been tested and validated. What are the shortcomings of practitioner-friendly models? In what contexts are they most useful? Are they as easy for practitioners to use as they appear to be? What challenges do practitioners face in their use? What resources would a practitioner need to implement a model such as the Fogg Behavior Model in an LMIC? This paper paves the way for a broader discussion on this topic by behavioral scientists working in low-resource settings.

## Figures and Tables

**Figure 1 vaccines-10-01261-f001:**
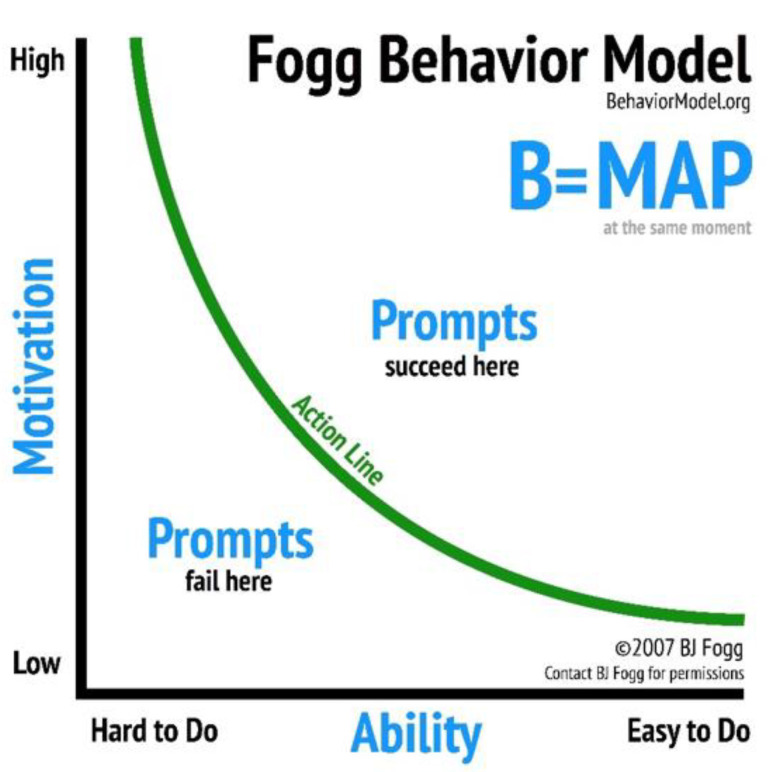
Fogg Behavior Model.

**Table 1 vaccines-10-01261-t001:** Pakistan survey questions related to the Fogg Behavior Model.

	Questions
Motivation	
Anticipation (hopes/fears)	On a scale from strongly agree to strongly disagree, how do you respond to the following: Family Planning can help improve one’s standard of living
Sensation (pleasure/pain)	On a scale from strongly agree to strongly disagree, how do you respond to the following: 2. Condom use reduces sexual satisfaction 3. Condom use reduces sexual desire 4. Use of condoms interrupts sex 5. I do not enjoy using condoms 6. Condoms cause skin irritation after sex 7. Condoms cause itching 8. Condoms have a bad smell
Belonging	On a scale from approve to disapprove, how do you respond to the following: 9. Would you say that you approve or disapprove of couples using a contraceptive method to avoid getting pregnant? 10. Would you say that *your wife* approves or disapproves of couples using a contraceptive method to avoid getting pregnant? On a scale from strongly agree to strongly disagree, how do you respond to the following: 11. Religion finds family planning acceptable 12. Men should share the responsibility for family planning 13. Child spacing protects the health of mothers 14. Spouses who care for one another will use family planning
Ability	
Time	None available
Money	Do you think the price for condoms is inexpensive, just affordable, or too expensive?
Physical effort	On a scale from strongly agree to strongly disagree, how do you respond to the following: 2. I know a place nearby where I can obtain condoms 3. Many different brands of condoms are available in this area 4. There is a general store in this neighborhood where I could obtain condoms 5. I can easily obtain condoms 6. There is a pharmacy nearby which sells condoms
Mental Effort	On a scale from strongly agree to strongly disagree, how do you respond to the following: 7. When I suggest using a condom, I am almost always embarrassed 8. It is really hard to bring up the issue of using condoms to my wife 9. I never know what to say when my wife and I need to talk about condoms or other protection 10. I am not comfortable talking about condoms with my wife 11. It is very embarrassing to buy condoms 12. When I need condoms, I often dread having to get them 13. It would be embarrassing to be seen buying condoms in a store 14. I always feel really uncomfortable when I buy condoms 15. Due to religious beliefs, a person feels guilty about using condoms
Routine	None available

**Table 2 vaccines-10-01261-t002:** Nigeria survey questions related to the Fogg Behavior Model.

	Questions
Motivation	
Anticipation (hopes/fears)	Please tell me if you strongly agree, somewhat agree, somewhat disagree, strongly disagree, or DK (don’t know): Condoms prevent pregnancy Condoms have holes that allow HIV to pass through them Contraceptives can cause cancer Contraceptives are dangerous to your health How motivated or unmotivated are you to use condoms with your partner? Very motivated, somewhat motivated, unmotivated, very unmotivated or don’t know How motivated or unmotivated are you to discuss contraception with your partner? Very motivated, somewhat motivated, unmotivated, very unmotivated or don’t know Do you intend to talk to your partner about contraception in the next three months?
Sensation (pleasure/pain)	Please tell me if you strongly agree, somewhat agree, somewhat disagree, strongly disagree, or DK: 8. Sex is unnatural with condoms 9. Condoms ruin the mood 10. Contraceptives reduce a woman’s sexual urge 11. Contraceptives reduce a man’s sexual urge
Belonging	Please tell me if you strongly agree, somewhat agree, somewhat disagree, strongly disagree, or DK: 12. It is against your values to have sexual intercourse before marriage 13. Adolescent girls who get pregnant before marriage should feel ashamed? 14. Adolescents should have sex before marriage to see if they are suited to each other 15. Condom use means that a person is promiscuous 16. Women who use contraceptives may become promiscuous s 17. Use of some contraceptives can make a woman permanently infertile 18. The husband should be the one to decide whether the couple should use a method of contraception 19. On a scale of 1–7 please tell me how having sex makes a person cool. 1 is not cool and 7 is cool. 20. On a scale of 1–7 please tell me how having sex makes a person sexy. 1 is not sexy and 7 is sexy 21. On a scale of 1–7 please tell me how having sex makes a person respected. 1 is not respected and 7 is respected 22. Have you and your partner ever discussed the number of children you would like to have?
Ability	
Time	None available
Money	None available
Physical effort	None available
Mental Effort	Please tell me if you strongly agree, somewhat agree, somewhat disagree, strongly disagree, or DK: Using condoms during sexual intercourse is wise Using condoms during sexual intercourse is embarrassing *Please tell me how easy or difficult it would be to do each of the following* 3. How easy or difficult would it be for you to use condoms with a sexual partner? 4. How easy or difficult would it be for you to discuss condoms with a sexual partner? 5 How easy or difficult would it be for you to discuss condoms with your parents? 6. How easy or difficult is it for you to use contraception? Please tell me how confident you would feel, extremely confident, somewhat confident, somewhat uncertain, extremely uncertain or DK: 7. How confident are you that you could get a condom if you wanted one? 8. How confident are you that you could have a condom with you when you needed it, that is if you decided to have sex? 9. How confident are you that you could use a condom correctly? 10. Imagine that you are having sex with someone you just met, and you feel it is important to use condoms. How confident are you that you could tell that person you want to use condoms? 11. Imagine that your partner uses birth control pills to prevent pregnancy. You want to use condoms to avoid getting an STD or HIV. How confident are you that you could convince your partner to also use condoms? 12. How confident are you that you could convince your partner to use a method of contraception? 13. How confident are you that you could use a method of contraception even if your partner doesn’t want to? Now I am going to ask you about the likelihood of some events. Please tell me if you would be extremely unlikely, somewhat unlikely, somewhat likely, extremely likely or don’t know. 14. How likely is it that your partner would like it if you had a condom with you? 15. During the next 3 months, how likely is it that you will try to persuade your partner to use condoms every time you have sex? 16. During the next 3 months, how likely is it that you will always have a condom with you? 17. Do you know of a place where you can obtain a method of contraception?
Routine	18. Do you carry condoms with you?

**Table 3 vaccines-10-01261-t003:** Frequency distributions, cross tabulations and the adjusted odds of an HCW getting at least one dose of a COVID-19 vaccine, Nigeria.

	(1) Frequency Distribution of Sample Characteristics (n = 496)	(2) % Who Got at Least one Dose of a COVID-19 Vaccine	(3) Logistic Regression Showing Odds of Getting One Dose of COVID-19 Vaccine aOR (95% CI)
**Motivation and ability**			
Low motivation and low ability	26.0	24.8%	1.00
High motivation or high ability	47.4%	56.2%	3.94 *** (2.33, 6.65)
High motivation and high ability	26.6%	89.4% ***	26.6 *** (12.86, 55.21)
**Region**			
North West	10.5%	75.0%	3.97 ** (1.59, 9.93)
North East	11.9%	64.4%	2.10 (0.90, 4.86)
North Central	19.8%	59.2%	2.23 * (1.07, 4.65)
South West	28.6%	55.6%	1.64 (0.83, 3.25)
South South	14.5%	38.9%	1.00
South East	14.7%	54.8%**	1.81 (0.82, 4.00)
**Age**			
18–29	30.4%	55.6%	1.00
30–39	51.2%	55.5%	1.01 (0.62, 1.64)
40 and older	18.3%	62.6%	0.91 (0.47, 1.76)
**Gender**			
Male	51.2%	61.4%	1.00
Female	48.8%	53.2%	0.92 (0.57, 1.45)
**Education**			
Up to secondary	16.6%	50.0%	1.00
Bachelor’s	41.3%	55.4%	1.20 (0.64, 2.24)
Diploma	28.9%	60.8%	1.24 (0.65, 2.37)
Master’s or higher	13.2%	61.5%	0.98 (0.43, 2.23)
**Type of healthcare worker (HCW)**			
Nurse, midwife, community health worker	84.1%	54.4%	1.00
Physician	15.9%	69.6% *	2.68 ** (1.39, 5.14)
**Is NPHCDA managing COVID-19 well?**			
Not at all or don’t know	19.2%	42.1%	1.00
Yes, somewhat	38.3%	55.8%	1.89 * (1.02, 3.49)
Yes, definitely	42.5%	64.5% **	1.95 * (1.06, 3.58)
Total	100%	56.9%	
Pseudo R^2^	-	-	22.20%
Number of HCWs	-	-	484

* *p* < 0.05, ** *p* < 0.01, *** *p* < 0.001 aOR = Adjusted Odds Ratios; 95%CI = 95% Confidence Interval.

**Table 4 vaccines-10-01261-t004:** Frequency distributions, cross tabulations and adjusted odds of condom use among men, Pakistan.

	(1) Frequency Distribution of Sample Characteristics (n = 617)	(2) % of Men Who Used a Condom at Last Sex	(3) Logistic Regression Showing the Adjusted Odds of Condom Use aOR (95% CI)
**Motivation and ability**			
Low motivation and low ability	55.9%	6.7%	1.00
High motivation or high ability	25.1%	25.2%	6.36 *** (3.24, 12.48)
High motivation and high ability	19.0%	44.4% ***	35.33 *** (15.01, 83.16)
**Age**			
<30	15.2%	13.8%	1.00
30–39	37.0%	24.6%	1.37 (0.63, 2.98)
40+	47.8%	15.2% *	0.77 (0.35, 1.70)
**Number of children**			
0–1	20.7%	9.4%	1.00
2–3	31.8%	23.0%	3.46 * (1.25, 9.56)
4+	47.5%	19.4% **	2.17 (0.76, 6.15)
**Desire for additional children**			
Want additional children	57.2%	13.0%	1.00
Want no additional children	42.8%	25.8% ***	3.32 *** (1.74, 6.33)
**Education**			
None or primary	29.7%	13.1%	1.00
Middle	16.9%	16.4%	1.37 (0.55, 3.43)
Grade 10 or higher	53.5%	22.1% *	1.31 (0.59, 2.87)
**Wealth**			
First four quintiles	80.4%	16.9%	1.00
Richest/fifth quintile	19.6%	24.8% *	2.35 * (1.19, 4.63)
**Advertising exposure**			
No exposure	84.8%	16.8%	1.00
Exposure to *Touch* condom advertising	15.2%	27.7% *	1.99 * (1.02, 3.87)
**Time**			
Baseline	-	-	1.00
Follow-up	-	-	1.32 (0.85, 2.05)
Total	100%	18.5%	
Number of men	-	-	617

* *p* < 0.05, ** *p* < 0.01, *** *p* < 0.001 aOR = Adjusted Odds Ratios; 95%CI = 95% Confidence Interval.

**Table 5 vaccines-10-01261-t005:** Frequency distributions, cross tabulations and adjusted odds of odds of iron folate use among women, India.

	(1) Frequency Distribution of Sample Characteristics (n = 1136)	(2) % of Women Who Use Iron Folate	(3) Logistic Regression Showing the Adjusted Odds of Iron Folate Use aOR (95% CI)
**Motivation and ability**			
Low motivation and low ability	53.3%	12.5%	1.00
High motivation or high ability	30.5%	27.7%	2.23 *** (1.57, 3.15)
High motivation and high ability	16.2%	63.6% ***	9.41 *** (6.31, 14.03)
**Age**			
Up to 25 years	54.7%	22.7%	1.00
26 and older	45.3%	28.7% *	1.70 *** (1.22, 2.38)
**Years of schooling completed**			
5 or more years	78.8%	23.5%	1.00
Less than 5 years	21.2%	32.8% **	1.61 * (1.08, 2.39)
**Exposure to advertising**			
Not exposed	83.7%	20.7%	1.00
Exposed	16.3%	49.7 ***	2.25 *** (1.55, 3.28)
Total	100%	25.4%	
Pseudo R-squared	-	-	16.31%
Number of women	-	-	1136

* *p* < 0.05, ** *p* < 0.01, *** *p* < 0.001 aOR = Adjusted Odds Ratios; 95%CI = 95% Confidence Interval.

**Table 6 vaccines-10-01261-t006:** Logistic regression showing the adjusted odds of modern contraceptive use among adolescent and young women, Nigeria.

	(1) Frequency Distribution of Sample Characteristics (n = 618)	(2) % of Women Who Currently Use Modern Contraception	(3) Logistic Regression Showing the Adjusted Odds of Modern Contraceptive Use aOR (95% CI)
**Motivation and ability**			
Low motivation and low ability	63.6	14.0	1.00
High motivation or high ability	23.0	43.7	3.93 *** (2.49, 6.23)
High motivation and high ability	13.4	61.4 ***	7.91 *** (4.38, 14.26)
**Region**			
South	34.1%	34.1%	1.00
North	65.9%	23.6% **	1.21 (0.80, 1.84)
**Age**			
14–19	26.2%	22.2%	1.00
20–24	73.4%	28.9%	1.25 (0.78, 2.02)
**Marital status**			
Married	64.1%	18.9%	1.00
Boyfriend	35.9%	41.9% ***	2.42 **(1.40, 4.20)
**Number of children**			
1–3	52.1%	21.4%	1.00
None	47.9%	33.4% **	1.04 (0.61, 1.79)
**Education**			
None or primary	19.4%	20.0%	1.00
Secondary	66.7%	25.0%	0.71 (0.43, 1.15)
Higher than secondary	13.9%	47.7% ***	1.23 (0.57, 2.64)
**Wealth**			
Second to fifth quintiles	79.9%	30.2%	1.00
First quintile/Poorest	20.1%	15.3% **	0.45 ** (0.26, 0.80)
Total	100%	27.2%	
Pseudo R^2^	-	-	17.90%
Number of women	-	-	618

* *p* < 0.05, ** *p* < 0.01, *** *p* < 0.001 aOR = Adjusted Odds Ratios; 95%CI = 95% Confidence Interval.

## Data Availability

The datasets used for the current study are available from the corresponding author on reasonable request.

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
