# Peer review of "Use of a Practitioner-Friendly Behavior Model to Identify Factors Associated with COVID-19 Vaccination and Other Behaviors"

_vaccines, 2022, doi:10.3390/vaccines10081261_

Round 1
Reviewer 1 Report
Abstract
- The abstract requires revision. Since the author wants to illustrate the efficacy of FBM to integrate into BCI, this is not a correct way to mention about COVID-19. Moreover, the draft examines four different behaviors which does not reflect in the background of the study by mentioning only COVID-19. This should be focused to make argument why FBM should be employed in the BCI.
- I would like to invite author to add some results in quantitative way while claiming the efficacy of FBM across the behaviors they studied.
Introduction
- Line 42- 54: The author made some critical points without citing any papers to support their arguments. Many BCIs are working well in different LMIC settings. I am unsure how the author got to know about these limitations without citing any references. So, I would like to invite author to cite relevant papers before making these arguments.
- Line 69-74: Again, the author mentioned the basics of FBM without citing it.
- Lines 42-54 and 69-74: As a reader I was expecting to have a critical understanding what makes it different and friendly compare to other approaches. This is the most important part to answer: why do practitioners should employ FBM while there are important approaches are working in BCIs.
- Figure 1: It says, the copy right belongs to the founder Prof Fogg. So, it requires permission to use the original illustration. If it is permitted, please give the source underneath the figure.
Methods
- Line 112-115: That’s an interesting point. Pakistan study was an effectiveness study, while the Nigeria study is formative in nature which is completely different because the BCIs were developed based on Social Marketing Approach. Theoretically, these are very different approaches.
- The methodology that was used in this study is less convincing. The author used a variety of behaviors to come with a conclusion on the effectiveness of the practitioner friendly approach. Interestingly, the India, Pakistan and Nigeria studies had different types of settings, questions, behaviors and year of data collection.
- While looking at the operationalization of the study approach, I do not find it methodologically correct as because, the author followed different approaches across all behaviors they studied. This should be uniform.
- Questionnaire, the author did not provide all the questionnaires that they used for data collection. It is really difficult to understand if all the domains were covered in all studies.
Discussion
- While reading the discussion section, I found there was not concrete analysis of the results.
- The author started the discussion with the current COVID-19 situation and the need of behavioral intervention to combat vaccine hesitancy and promote vaccine acceptance. The next point they made is the suitability of the approach. The discussion points were not supported by the current literature. There are no argument or comparison with other approaches.
- Finally, author claimed a number of suitability of the FBM method. However, without having tested this approach, it is really difficult to come a conclusion if this approach would work well or not.
Final opinion:
Given the differences in study behaviors, settings, analytical approaches and year of data collection, this paper should not be accepted in its current form. Rather, it could be considered as an opinion piece, and that need further revisions in the methodology section. The author can create case stories by utilizing the results of each behavior in each country and come up with a recommendation that FBM approach could be a potential solution in response to the current limitations in the BCIs.
Author Response
We are grateful to the reviewer for their comments.
Please see the attachment for a point-by-point response to the reviewer's comments.

Reviewer 2 Report
The implicit measurement of motivation and ability is inappropriate. The FOGG model is not recent.. It was published in 2009. This paper does not contribute to what is known in the social and behavioral sciences literature about theory-based interventions. The multiple ways that motivation and ability is measured is not comparable across studies. Where multiple items are used to measure the variables and then one-item is used to measure the two variables (motivation and ability) is not good science.
Author Response

(The authors gave the same response as above.)

Reviewer 3 Report
The proposed paper is devoted to the application of the Fogg Behavior Model to the identification of important factors related to COVID-19 Vaccination, use of contraceptives and other behaviors.
The author study the problem whether an individual with high motivation and high ability is significantly more likely to adopt a behavior than one with lower levels of motivation and ability.
Preliminaries to the research area are provided, including an explanation of the need for development and application of “practitioner-friendly” models of behavior, i.e. models which can be easily used by intervention designers, implementers, and evaluators with limited technical and financial resources. The main ideas of the Fogg Behavior Model are described.
Details about the survey data from Nigeria, Pakistan and India are given. The are related to the motivation and ability for uptake of COVID-19 vaccine, use of condoms, iron folate and modern contraceptives.
Details about the operationalization of the Fogg Behavior Model are provided. The variables included into the models are described and their interrelations are studied.
The limitations of the study are described.
Results of the statistical analysis are presented. They are analyzed and commented in details.
Interesting research perspectives are provided.
The presentation of the main results is clear. From a formal point of view, all the contents seems to be correct. The results are valuable and worthy of being published taking into account their possible applications in various areas of medicine and public health, behavior change theories, especially for behavioral scientists working in low-resource settings.
Minor revisions are suggested to improve the quality of the exposition:
p. 4, line 134: I suggest authors to write “using data about” instead of “using data on”.
p. 7, line 225: I suggest the author to explain the meaning of the abbreviation „DK” in Table 2.
Author Response

(The authors gave the same response as above.)

Round 2
Reviewer 2 Report
I appreciate the rationale for the manuscript and its focus on social and behavioral sciences. It is well needed in public health practice.
The instruments used to measure the variables, motivation, and ability, were not explicitly designed to measure cause and ability. Therefore, these are implicit. The authors "fit" the instruments to the variables rather than explicit measures of motivation and ability. Most behavioral science interventions are not explicitly based on theoretical variables. We work backward to fit the variables into a theory/model. The instruments were not direct measures of motivation and ability.
I still believe that a one-item instrument to measure a theoretical variable is not good science. I am unfamiliar with any expert supporting one-item instrument to measure a theoretical variable. The authors should note this limitation.
I understand this is not the forum for a discussion on the value of the FOGG Model compared to other models or theories. Perhaps the FOGG Model does not lend itself to intervention development, while other theories (social cognitive theory) may lend themselves well to behavioral intervention development. When operationalizing SCT, it shows it is skills-based.
The revisions the authors made to the manuscript are valuable.
Author Response
The reviewer’s point is valid. We have addressed this in the methodology section.
We used global, single-item, scales of motivation and ability in two of the studies: the study with HCWs in Nigeria and the study with women of reproductive ages in India. The use of multi-item scales is recommended in the literature and widely practiced by researchers. While there are concerns regarding measurement error resulting from single-item scales, several studies have shown that single-item scales perform as well or better for topics such as job satisfaction [28] or perform reasonably well for topics such as quality of life [29] or depression [30]. More research is needed to determine how well single-item measures of motivation and ability perform against multi-item scales.
- Scarpello V, Campbell JP. Job Satisfaction: are all the parts there? Personnel Psychology. 2006; 36(3):577-600.
- Cunny KA, Perri M. Single-item vs multiple-item measures of health-related quality of life. Psychol Rep. 1991;69(1):127-30.
- McKenzie N, Marks I. Quick rating of depressed mood in patients with anxiety disorders. Br J Psychiatry. 1999; 174:266-9.
Sincerely,
Sohail Agha